# Expressing an Image Stream with a Sequence of Natural Sentences

**Cesc Chunseong Park**     **Gunhee Kim**
Seoul National University, Seoul, Korea
{park.chunseong,gunhee}@snu.ac.kr
https://github.com/cesc-park/CRCN

## Abstract

We propose an approach for retrieving a sequence of natural sentences for an image stream. Since general users often take a series of pictures on their special moments, it would better take into consideration of the whole image stream to produce natural language descriptions. While almost all previous studies have dealt with the relation between a single image and a single natural sentence, our work extends both input and output dimension to a sequence of images and a sequence of sentences. To this end, we design a multimodal architecture called *coherent recurrent convolutional network* (CRCN), which consists of convolutional neural networks, bidirectional recurrent neural networks, and an entity-based local coherence model. Our approach directly learns from vast user-generated resource of blog posts as text-image parallel training data. We demonstrate that our approach outperforms other state-of-the-art candidate methods, using both quantitative measures (*e.g.* BLEU and top-K recall) and user studies via Amazon Mechanical Turk.

## 1   Introduction

Recently there has been a hike of interest in automatically generating natural language descriptions for images in the research of computer vision, natural language processing, and machine learning (*e.g.* [5, 8, 9, 12, 14, 15, 26, 21, 30]). While most of existing work aims at discovering the relation between a single image and a single natural sentence, we extend both input and output dimension to a *sequence* of images and a *sequence* of sentences, which may be an obvious next step toward joint understanding of the visual content of images and language descriptions, albeit under-addressed in current literature. Our problem setup is motivated by that general users often take a series of pictures on their memorable moments. For example, many people who visit *New York City* (NYC) would capture their experiences with large image streams, and thus it would better take the whole photo stream into consideration for the translation to a natural language description.

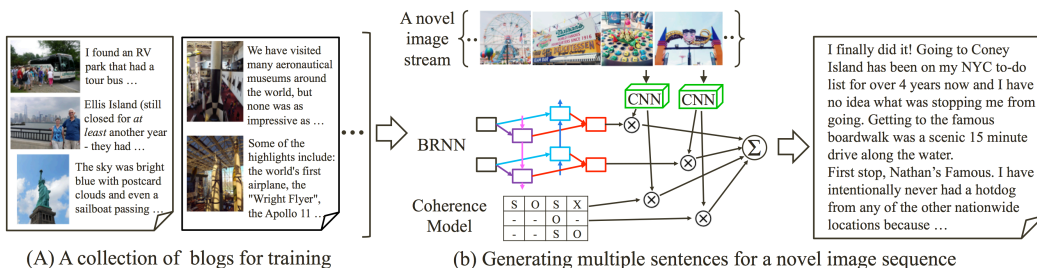

(A) A collection of blogs for training          (b) Generating multiple sentences for a novel image sequence

Figure 1: An intuition of our problem statement with a *New York City* example. We aim at expressing an image stream with a sequence of natural sentences. (a) We leverage natural blog posts to learn the relation between image streams and sentence sequences. (b) We propose *coherent recurrent convolutional networks* (CRCN) that integrate convolutional networks, bidirectional recurrent networks, and the entity-based coherence model.

Fig.1 illustrates an intuition of our problem statement with an example of *visiting NYC*. Our objective is, given a photo stream, to automatically produce a sequence of natural language sentences that best describe the essence of the input image set. We propose a novel multimodal architecture named *coherent recurrent convolutional networks* (CRCN) that integrate convolutional neural networks for image description [13], bidirectional recurrent neural networks for the language model [20], and the local coherence model [1] for a smooth flow of multiple sentences. Since our problem deals with learning the semantic relations between long streams of images and text, it is more challenging to obtain appropriate text-image parallel corpus than previous research of single sentence generation. Our idea to this issue is to directly leverage online natural blog posts as text-image parallel training data, because usually a blog consists of a sequence of informative text and multiple representative images that are carefully selected by authors in a way of storytelling. See an example in Fig.1.(a).

We evaluate our approach with the blog datasets of the *NYC* and *Disneyland*, consisting of more than 20K blog posts with 140K associated images. Although we focus on the tourism topics in our experiments, our approach is completely unsupervised and thus applicable to any domain that has a large set of blog posts with images. We demonstrate the superior performance of our approach by comparing with other state-of-the-art alternatives, including [9, 12, 21]. We evaluate with quantitative measures (*e.g.* BLEU and Top-K recall) and user studies via Amazon Mechanical Turk (AMT).

**Related work**. Due to a recent surge of volume of literature on this subject of generating natural language descriptions for image data, here we discuss a representative selection of ideas that are closely related to our work. One of the most popular approaches is to pose the text generation as a retrieval problem that learns ranking and embedding, in which the caption of a test image is transferred from the sentences of its most similar training images [6, 8, 21, 26]. Our approach partly involves the text retrieval, because we search for candidate sentences for each image of a query sequence from training database. However, we then create a final paragraph by considering both compatibilities between individual images and text, and the coherence that captures text relatedness at the level of sentence-to-sentence transitions. There have been also video-sentence works (*e.g.* [23, 32]); our key novelty is that we explicitly include the coherence model. Unlike videos, consecutive images in the streams may show sharp changes of visual content, which cause the abrupt discontinuity between consecutive sentences. Thus the coherence model is more demanded to make output passages fluent.

Many recent works have exploited multimodal networks that combine deep convolutional neural networks (CNN) [13] and recurrent neural network (RNN) [20]. Notable architectures in this category integrate the CNN with bidirectional RNNs [9], long-term recurrent convolutional nets [5], long-short term memory nets [30], deep Boltzmann machines [27], dependency-tree RNN [26], and other variants of multimodal RNNs [3, 19]. Although our method partly take advantage of such recent progress of multimodal neural networks, our major novelty is that we integrate it with the coherence model as a unified end-to-end architecture to retrieve fluent sequential multiple sentences.

In the following, we compare more previous work that bears a particular resemblance to ours. Among multimodal neural network models, the long-term recurrent convolutional net [5] is related to our objective because their framework explicitly models the relations between sequential inputs and outputs. However, the model is applied to a video description task of creating a sentence for a given short video clip and does not address the generation of multiple sequential sentences. Hence, unlike ours, there is no mechanism for the coherence between sentences. The work of [11] addresses the retrieval of image sequences for a query paragraph, which is the opposite direction of our problem. They propose a latent structural SVM framework to learn the semantic relevance relations from text to image sequences. However, their model is specialized only for the image sequence retrieval, and thus not applicable to the natural sentence generation.

**Contributions**. We highlight main contributions of this paper as follows. (1) To the best of our knowledge, this work is the first to address the problem of expressing image streams with sentence sequences. We extend both input and output to more elaborate forms with respect to a whole body of existing methods: image streams instead of individual images and sentence sequences instead of individual sentences. (2) We develop a multimodal architecture of *coherent recurrent convolutional networks* (CRCN), which integrates convolutional networks for image representation, recurrent networks for sentence modeling, and the local coherence model for fluent transitions of sentences. (3) We evaluate our method with large datasets of unstructured blog posts, consisting of 20K blog posts with 140K associated images. With both quantitative evaluation and user studies, we show that our approach is more successful than other state-of-the-art alternatives in verbalizing an image stream.

## 2 Text-Image Parallel Dataset from Blog Posts

We discuss how to transform blog posts to a training set $\mathcal{B}$ of image-text parallel data streams, each of which is a sequence of image-sentence pairs: $B^l = \{(I_1^l, T_1^l), \cdots, (I_{N^l}^l, T_{N^l}^l)\} \in \mathcal{B}$. The training set size is denoted by $L = |\mathcal{B}|$. Fig.2.(a) shows the summary of pre-processing steps for blog posts.

### 2.1 Blog Pre-processing

We assume that blog authors augment their text with multiple images in a semantically meaningful manner. In order to decompose each blog into a sequence of images and associated text, we first perform *text segmentation* and then *text summarization*. The purpose of text segmentation is to divide the input blog text into a set of text segments, each of which is associated with a single image. Thus, the number of segments is identical to the number of images in the blog. The objective of text summarization is to reduce each text segment into a single key sentence. As a result of these two processes, we can transform each blog into a form of $B^l = \{(I_1^l, \mathcal{T}_1^l), \cdots, (I_{N^l}^l, \mathcal{T}_{N^l}^l)\}$.

**Text segmentation**. We first divide the blog passage into text blocks according to paragraphs. We apply a standard paragraph tokenizer of NLTK [2] that uses rule-based regular expressions to detect paragraph divisions. We then use the heuristics based on the image-to-text block distances proposed in [10]. Simply, we assign each text block to the image that has the minimum index distance where each text block and image is counted as a single index distance in the blog.

**Text summarization**. We summarize each text segment into a single key sentence. We apply the *Latent Semantic Analysis* (LSA)-based summarization method [4], which uses the singular value decomposition to obtain the concept dimension of sentences, and then recursively finds the most representative sentences that maximize the inter-sentence similarity for each topic in a text segment.

**Data augmentation**. The data augmentation is a well-known technique for *convolutional neural networks* to improve image classification accuracies [13]. Its basic idea is to artificially increase the number of training examples by applying transformations, horizontal reflection or adding noise to training images. We empirically observe that this idea leads better performance in our problem as well. For each image-sentence sequence $B^l = \{(I_1^l, T_1^l), \cdots, (I_{N^l}^l, T_{N^l}^l)\}$, we augment each sentence $T_n^l$ with multiple sentences for training. That is, when we perform the LSA-based text summarization, we select top-$\kappa$ highest ranked summary sentences, among which the top-ranked one becomes the summary sentence for the associated image, and all the top-$\kappa$ ones are used for training in our model. With a slight abuse of notation, we let $T_n^l$ to denote both the single summary sentence and $\kappa$ augmented sentences. We choose $\kappa = 3$ after thorough empirical tests.

### 2.2 Text Description

Once we represent each text segment with $\kappa$ sentences, we extract the *paragraph vector* [17] to represent the content of text. The *paragraph vector* is a neural-network based unsupervised algorithm that learns fixed-length feature representation from variable-length pieces of passage. We learn 300-dimensional dense vector representation separately from the two classes of the blog dataset using the `gensim doc2vec` code. We use $p_n$ to denote the paragraph vector representation for text $T_n$. We then extract a parsed tree for each $T_n$ to identify coreferent entities and grammatical roles of the words. We use the Stanford core NLP library [18]. The parse trees are used for the local coherence model, which will be discussed in section 3.2.

## 3 Our Architecture

Many existing sentence generation models (*e.g.* [9, 19]) combine words or phrases from training data to generate a sentence for a novel image. Our approach is one level higher; we use sentences from training database to author a sequence of sentences for a novel image stream. Although our model can be easily extended to use words or phrases as basic building blocks, such granularity makes sequences too long to train the language model, which may cause several difficulties for learning the RNN models. For example, the *vanishing gradient effect* is a well-known hardship to backpropagate an error signal through a long-range temporal interval. Therefore, we design our approach that retrieves individual candidate sentences for each query image from training database and crafts a best sentence sequence, considering both the fitness of individual image-to-sentence pairs and coherence between consecutive sentences.

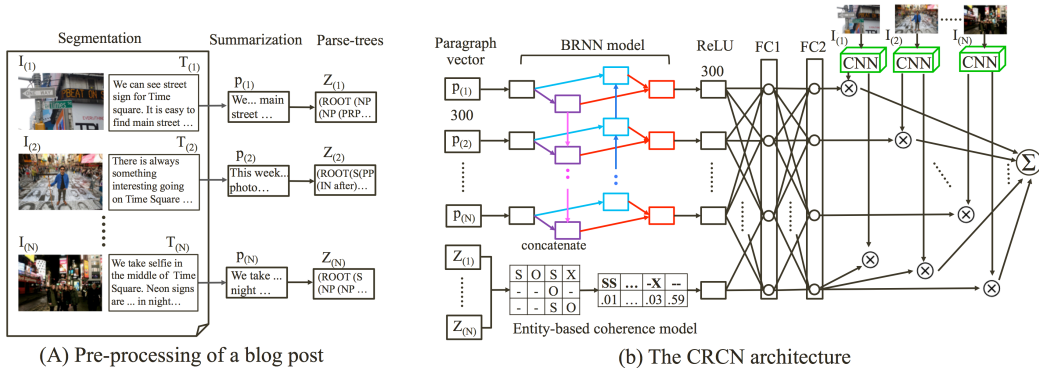

(A) Pre-processing of a blog post        (b) The CRCN architecture

Figure 2: Illustration of (a) pre-processing steps of blog posts, and (b) the proposed CRCN architecture.

Fig.2.(b) illustrates the structure of our CRCN. It consists of three main components, which are convolutional neural networks (CNN) [13] for image representation, bidirectional recurrent neural networks (BRNN) [24] for sentence sequence modeling, and the local coherence model [1] for a smooth flow of multiple sentences. Each data stream is a variable-length sequence denoted by $\{(I_1, T_1), \cdots, (I_N, T_N)\}$. We use $t \in \{1, \cdots, N\}$ to denote a position of a sentence/image in a sequence. We define the CNN and BRNN model for each position separately, and the coherent model for a whole data stream. For the CNN component, our choice is the VGGNet [25] that represents images as 4,096-dimensional vectors. We discuss the details of our BRNN and coherence model in section 3.1 and section 3.2 respectively, and finally present how to combine the output of the three components to create a single compatibility score in section 3.3.

## 3.1 The BRNN Model

The role of BRNN model is to represent a content flow of text sequences. In our problem, the BRNN is more suitable than the normal RNN, because the BRNN can simultaneously model forward and backward streams, which allow us to consider both previous and next sentences for each sentence to make the content of a whole sequence interact with one another. As shown in Fig.2.(b), our BRNN has five layers: input layer, forward/backward layer, output layer, and ReLU activation layer, which are finally merged with that of the coherent model into two fully connected layers. Note that each text is represented by 300-dimensional *paragraph vector* $p_t$ as discussed in section 2.2. The exact form of our BRNN is as follows. See Fig.2.(b) together for better understanding.

$$x_t^f = f(W_i^f p_t + b_i^f); \qquad x_t^b = f(W_i^b p_t + b_i^b); \tag{1}$$
$$h_t^f = f(x_t^f + W_f h_{t-1}^f + b_f); \; h_t^b = f(x_t^b + W_b h_{t+1}^b + b_b); \; o_t = W_o(h_t^f + h_t^b) + b_o.$$

The BRNN takes a sequence of text vectors $p_t$ as input. We then compute $x_t^f$ and $x_t^b$, which are the activations of input units to forward and backward units. Unlike other BRNN models, we separate the input activation into forward and backward ones with different sets of parameters $W_i^f$ and $W_i^b$, which empirically leads a better performance. We set the activation function $f$ to the Rectified Linear Unit (ReLU), $f(x) = \max(0, x)$. Then, we create two independent forward and backward hidden units, denoted by $h_t^f$ and $h_t^b$. The final activation of the BRNN $o_t$ can be regarded as a description for the content of the sentence at location $t$, which also implicitly encodes the flow of the sentence and its surrounding context in the sequence. The parameter sets to learn include weights $\{W_i^f, W_i^b, W_f, W_b, W_o\} \in \mathbb{R}^{300 \times 300}$ and biases $\{b_i^f, b_i^b, b_f, b_b, b_o\} \in \mathbb{R}^{300 \times 1}$.

## 3.2 The Local Coherence Model

The BRNN model can capture the flow of text content, but it lacks learning the coherence of passage that reflects distributional, syntactic, and referential information between discourse entities. Thus, we explicitly include a local coherence model based on the work of [1], which focuses on resolving the patterns of local transitions of discourse entities (*i.e.* coreferent noun phrases) in the whole text. As shown in Fig.2.(b), we first extract parse trees for every summarized text denoted by $Z_t$ and then concatenate all sequenced parse trees into one large one, from which we make an *entity grid* for the whole sequence. The entity grid is a table where each row corresponds to a discourse

entity and each column represents a sentence. Grammatical role are expressed by three categories and one for absent (*i.e.* not referenced in the sentence): **S** (subjects), **O** (objects), **X** (other than subject or object) and $-$(absent). After making the entity grid, we enumerate the transitions of the grammatical roles of entities in the whole text. We set the *history* parameter to three, which means we can obtain $4^3 = 64$ transition descriptions (*e.g.* **SO**$-$ or **OOX**). By computing the ratio of the occurrence frequency of each transition, we finally create a 64-dimensional representation that captures the coherence of a sequence. Finally, we make this descriptor to a 300-dimensional vector by zero-padding, and forward it to ReLU layer as done for the BRNN output.

### 3.3  Combination of CNN, RNN, and Coherence Model

After the ReLU activation layers of the RNN and the coherence model, their output (*i.e.* $\{o_t\}_{t=1}^N$ and $q$) goes through two fully connected (FC) layers, whose role is to decide a proper combination of the BRNN language factors and the coherence factors. We drop the bias terms for the fully-connected layers, and the dimensions of variables are $W_{f1} \in \mathbb{R}^{512 \times 300}, W_{f2} \in \mathbb{R}^{4,096 \times 512}$ , $o_t, q \in \mathbb{R}^{300 \times 1}$ , $s_t, g \in \mathbb{R}^{4,096 \times 1}, O \in \mathbb{R}^{300 \times N}$, and $S \in \mathbb{R}^{4,096 \times N}$.

$$O = [o_1|o_2|..|o_N]; \quad S = [s_1|s_2|..|s_N]; \quad W_{f2}W_{f1}[O|q] = [S|g]. \tag{2}$$

We use the shared parameters for $O$ and $q$ so that the output mixes well the interaction between the content flows and coherency. In our tests, joint learning outperforms learning the two terms with separate parameters. Note that the multiplication $W_{f2}W_{f1}$ of the last two FC layers does not reduce to a single linear mapping, thanks to dropout. We assign 0.5 and 0.7 dropout rates to the two layers. Empirically, it improves generalization performance much over a single FC layer with dropout.

### 3.4  Training the CRCN

To train our CRCN model, we first define the compatibility score between an image stream and a paragraph sequence. While our score function is inspired by Karpathy *et al.* [9], there are two major differences. First, the score function of [9] deals between sentence fragments and image fragments, and thus the algorithm considers all combinations between them to find out the best matching. On the other hand, we define the score by an ordered and paired compatibility between a sentence sequence and an image sequence. Second, we also add the term that measures the relevance relation of coherency between an image sequence and a text sequence. Finally, the score $S_{kl}$ for a sentence sequence $k$ and an image stream $l$ is defined by

$$S_{kl} = \sum_{t=1...N} s_t^k \cdot v_t^l + g^k \cdot v_t^l \tag{3}$$

where $v_t^l$ denotes the CNN feature vector for $t$-th image of stream $l$. We then define the cost function to train our CRCN model as follows [9].

$$C(\theta) = \sum_k \Big[ \sum_l \max(0, 1 + S_{kl} - S_{kk}) + \sum_l \max(0, 1 + S_{lk} - S_{kk}) \Big], \tag{4}$$

where $S_{kk}$ denotes the score between a training pair of corresponding image and sentence sequence. The objective, based on the max-margin structured loss, encourages aligned image-sentence sequence pairs to have a higher score by a margin than misaligned pairs. For each positive training example, we randomly sample 100 ne examples from the training set. Since each contrastive example has a random length, and is sampled from the dataset of a wide range of content, it is extremely unlikely that the negative examples have the same length and the same content order of sentences with positive examples.

**Optimization**. We use the backpropagation through time (BPTT) algorithm [31] to train our model. We apply the stochastic gradient descent (SGD) with mini-batches of 100 data streams. Among many SGD techniques, we select RMSprop optimizer [28], which leads the best performance in our experiments. We initialize the weights of our CRCN model using the method of He et al. [7], which is robust in deep rectified models. We observe that it is better than a simple Gaussian random initialization, although our model is not extremely deep. We use dropout regularization in all layers except the BRNN, with 0.7 dropout for the last FC layer and 0.5 for the other remaining layers.

## 3.5 Retrieval of Sentence Sequences

At test time, the objective is to retrieve a best sentence sequence for a given query image stream $\{I_{q1}, \cdots, I_{qN}\}$. First, we select $K$-nearest images for each query image from training database using the $\ell_2$-distance on the CNN VGGNet fc7 features [25]. In our experiments $K = 5$ is successful. We then generate a set of sentence sequence candidates $\mathcal{C}$ by concatenating the sentences associated with the $K$-nearest images at each location $t$. Finally, we use our learned CRCN model to compute the compatibility score between the query image stream and each sequence candidate, according to which we rank the candidates.

However, one major difficulty of this scenario is that there are exponentially many candidates (*i.e.* $|\mathcal{C}| = K^N$). To resolve this issue, we use an approximate divide-and-conquer strategy; we recursively halve the problem into subproblems, until the size of the subproblem is manageable. For example, if we halve the search candidate length $Q$ times, then the search space of each subproblem becomes $K^{N/2^Q}$. Using the beam search idea, we first find the top-$M$ best sequence candidates in the subproblem of the lowest level, and recursively increase the candidate lengths while the maximum candidate size is limited to $M$. We set $M = 50$. Though it is an approximate search, our experiments assure that it achieves *almost optimal* solutions with plausible combinatorial search, mainly because the *local fluency and coherence* is undoubtedly necessary for the *global one*. That is, in order for a whole sentence sequence to be fluent and coherent, its any subparts must be as well.

## 4 Experiments

We compare the performance of our approach with other state-of-the-art candidate methods via quantitative measures and user studies using Amazon Mechanical Turk (AMT). Please refer to the supplementary material for more results and the details of implementation and experimental setting.

### 4.1 Experimental Setting

**Dataset**. We collect blog datasets of the two topics: *NYC* and *Disneyland*. We reuse the blog data of Disneyland from the dataset of [11], and newly collect the data of *NYC*, using the same crawling method with [11], in which we first crawl blog posts and their associated pictures from two popular blog publishing sites, BLOGSPOT and WORDPRESS by changing query terms from Google search. Then, we manually select the travelogue posts that describe stories and events with multiple images. Finally, the dataset includes 11,863 unique blog posts and 78,467 images for *NYC* and 7,717 blog posts and 60,545 images for *Disneyland*.

**Task**. For quantitative evaluation, we randomly split our dataset into 80% as a training set, 10% as a validation, and the others as a test set. For each test post, we use the image sequence as a query $\mathcal{I}_q$ and the sequence of summarized sentences as groundtruth $\mathcal{T}_G$. Each algorithm retrieves the best sequences from training database for a query image sequence, and ideally the retrieved sequences match well with $\mathcal{T}_G$. Since the training and test data are disjoint, each algorithm can only retrieve similar (but not identical) sentences at best.

For quantitative measures, we exploit two types of metrics of language similarity (*i.e.* BLEU [22], CIDEr [29], and METEOR [16] scores) and retrieval accuracies (*i.e.* top-K recall and median rank), which are popularly used in text generation literature [8, 9, 19, 26]. The top-K recall R@K is the recall rate of a groundtruth retrieval given top K candidates, and the median rank indicates the median ranking value of the first retrieved groundtruth. A better performance is indicated by higher BLEU, CIDEr, METEOR, R@K scores, and lower median rank values.

**Baselines**. Since the sentence sequence generation from image streams has not been addressed yet in previous research, we instead extend several state-of-the-art single-sentence models that have publicly available codes as baselines, including the log-bilinear multimodal models by Kiros *et al.* [12], and recurrent convolutional models by Karpathy *et al.* [9] and Vinyals *et al.* [30]. For [12], we use the three variants introduced in the paper, which are the standard log-bilinear model (LBL), and two multi-modal extensions: modality-based LBL (MLBL-B) and factored three-way LBL (MLBL-F). We use the NeuralTalk package authored by Karpathy *et al.* for the baseline of [9] denoted by (CNN+RNN), and [30] denoted by (CNN+LSTM). As the simplest baseline, we also compare with the global matching (GloMatch) in [21]. For all the baselines, we create final sentence sequences by concatenating the sentences generated for each image in the query stream.

| | Language metrics | | | | | | Retrieval metrics | | | |
|---|---|---|---|---|---|---|---|---|---|---|
| | B-1 | B-2 | B-3 | B-4 | CIDEr | METEOR | R@1 | R@5 | R@10 | MedRank |
| **New York City** | | | | | | | | | | |
| (CNN+LSTM) [30] | 16.24 | 5.79 | 1.38 | 0.10 | 9.1 | 5.73 | 0.95 | 7.38 | 13.33 | 88.5 |
| (CNN+RNN) [9] | 6.21 | 0.01 | 0.00 | 0.00 | 0.5 | 1.34 | 0.48 | 2.86 | 4.29 | 120.5 |
| (MLBL-F) [12] | 21.03 | 1.92 | 0.12 | 0.01 | 4.3 | 6.03 | 0.71 | 4.52 | 7.86 | 87.0 |
| (MLBL-B) [12] | 20.43 | 1.54 | 0.09 | 0.01 | 2.6 | 5.30 | 0.48 | 3.57 | 5.48 | 101.5 |
| (LBL) [12] | 20.96 | 1.68 | 0.08 | 0.01 | 2.6 | 5.29 | 1.19 | 4.52 | 7.38 | 100.5 |
| (GloMatch) [21] | 19.00 | 1.59 | 0.04 | 0.0 | 2.80 | 5.17 | 0.24 | 2.62 | 4.05 | 95.00 |
| (1NN) | 25.97 | 3.42 | 0.60 | 0.22 | 15.9 | 7.06 | 5.95 | 13.57 | 20.71 | 63.50 |
| (RCN) | **27.09** | **5.45** | 2.56 | **2.10** | **33.5** | **7.87** | 3.80 | 18.33 | 30.24 | 29.00 |
| (CRCN) | 26.83 | 5.37 | **2.57** | 2.08 | 30.9 | 7.69 | **11.67** | **31.19** | **43.57** | **14.00** |
| **Disneyland** | | | | | | | | | | |
| (CNN+LSTM) [30] | 13.22 | 1.56 | 0.40 | 0.07 | 10.0 | 4.51 | 2.83 | 10.38 | 16.98 | 61.5 |
| (CNN+RNN) [9] | 6.04 | 0.00 | 0.00 | 0.00 | 0.4 | 1.34 | 1.02 | 3.40 | 5.78 | 88.0 |
| (MLBL-F) [12] | 15.75 | 1.61 | 0.07 | 0.01 | 4.9 | 7.12 | 0.68 | 4.08 | 10.54 | 63.0 |
| (MLBL-B) [12] | 15.65 | 1.32 | 0.05 | 0.00 | 3.8 | 5.83 | 0.34 | 2.72 | 6.80 | 69.0 |
| (LBL) [12] | 18.94 | 1.70 | 0.06 | 0.01 | 3.4 | 4.99 | 1.02 | 4.08 | 7.82 | 62.0 |
| (GloMatch) [21] | 11.94 | 0.37 | 0.01 | 0.00 | 2.2 | 4.31 | 2.04 | 5.78 | 7.48 | 73.0 |
| (1NN) | 25.92 | 3.34 | 0.71 | 0.38 | 19.5 | 7.46 | 9.18 | 19.05 | 27.21 | 45.0 |
| (RCN) | 28.15 | 6.84 | **4.11** | **3.52** | 51.3 | **8.87** | 5.10 | 20.07 | 28.57 | 29.5 |
| (CRCN) | **28.40** | **6.88** | **4.11** | 3.49 | **52.7** | 8.78 | **14.29** | **31.29** | **43.20** | **16.0** |

Table 1: Evaluation of sentence generation for the two datasets, *New York City* and *Disneyland*, with language similarity metrics (BLEU) and retrieval metrics (R@K, median Rank). A better performance is indicated by higher BLEU, CIDEr, METEOR, R@K scores, and lower median rank values.

We also compare between different variants of our method to validate the contributions of key components of our method. We test the $K$-nearest search (1NN) without the RNN part as the simplest variant; for each image in a test query, we find its $K(=1)$ most similar training images and simply concatenate their associated sentences. The second variant is the BRNN-only method denoted by (RCN) that excludes the entity-based coherence model from our approach. Our complete method is denoted by (CRCN), and this comparison quantifies the improvement by the coherence model. To be fair, we use the same VGGNet fc7 feature [25] for all the algorithms.

## 4.2 Quantitative Results

Table 1 shows the quantitative results of experiments using both language and retrieval metrics. Our approach (CRCN) and (RCN) outperform, with large margins, other state-of-the-art baselines, which generate passages without consideration of sentence-to-sentence transitions unlike ours. The (MLBL-F) shows the best performance among the three models of [12] albeit with a small margin, partly because they share the same word dictionary in training. Among mRNN-based models, the (CNN+LSTM) significantly outperforms the (CNN+RNN), because the LSTM units help learn models from irregular and lengthy data of natural blogs more robustly.

We also observe that (CRCN) outperforms (1NN) and (RCN), especially with the retrieval metrics. It shows that the integration of two key components, the BRNN and the coherence model, indeed contributes the performance improvement. The (CRCN) is only slightly better than the (RCN) in language metrics but significantly better in retrieval metrics. It means that (RCN) is fine with retrieving fairly good solutions, but not good at ranking the only correct solution high compared to (CRCN). The small margins in language metrics are also attributed by their inherent limitation; for example, the BLEU focuses on counting the matches of $n$-gram words and thus is not good at comparing between sentences, even worse between paragraphs for fully evaluating their fluency and coherency.

Fig.3 illustrates several examples of sentence sequence retrieval. In each set, we show a query image stream and text results created by our method and baselines. Except Fig.3.(d), we show parts of sequences because they are rather long for illustration. These qualitative examples demonstrate that our approach is more successful to verbalize image sequences that include a variety of content.

## 4.3 User Studies via Amazon Mechanical Turk

We perform user studies using AMT to observe general users' preferences between text sequences by different algorithms. Since our evaluation involves multiple images and long passages of text, we design our AMT task to be sufficiently simple for general turkers with no background knowledge.

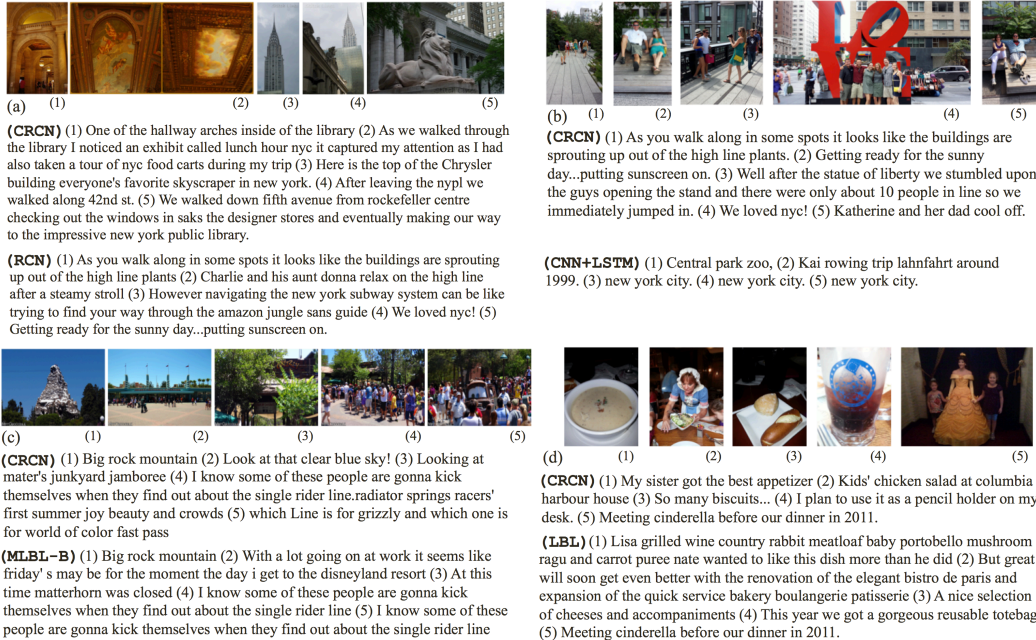

**(CRCN)** (1) One of the hallway arches inside of the library (2) As we walked through the library I noticed an exhibit called lunch hour nyc it captured my attention as I had also taken a tour of nyc food carts during my trip (3) Here is the top of the Chrysler building everyone's favorite skyscraper in new york. (4) After leaving the nypl we walked along 42nd st. (5) We walked down fifth avenue from rockefeller centre checking out the windows in saks the designer stores and eventually making our way to the impressive new york public library.

**(RCN)** (1) As you walk along in some spots it looks like the buildings are sprouting up out of the high line plants (2) Charlie and his aunt donna relax on the high line after a steamy stroll (3) However navigating the new york subway system can be like trying to find your way through the amazon jungle sans guide (4) We loved nyc! (5) Getting ready for the sunny day...putting sunscreen on.

**(CRCN)** (1) As you walk along in some spots it looks like the buildings are sprouting up out of the high line plants. (2) Getting ready for the sunny day...putting sunscreen on. (3) Well after the statue of liberty we stumbled upon the guys opening the stand and there were only about 10 people in line so we immediately jumped in. (4) We loved nyc! (5) Katherine and her dad cool off.

**(CNN+LSTM)** (1) Central park zoo, (2) Kai rowing trip lahnfahrt around 1999. (3) new york city. (4) new york city. (5) new york city.

**(CRCN)** (1) Big rock mountain (2) Look at that clear blue sky! (3) Looking at mater's junkyard jamboree (4) I know some of these people are gonna kick themselves when they find out about the single rider line.radiator springs racers' first summer joy beauty and crowds (5) which Line is for grizzly and which one is for world of color fast pass

**(MLBL-B)** (1) Big rock mountain (2) With a lot going on at work it seems like friday' s may be for the moment the day i get to the disneyland resort (3) At this time matterhorn was closed (4) I know some of these people are gonna kick themselves when they find out about the single rider line (5) I know some of these people are gonna kick themselves when they find out about the single rider line

**(CRCN)** (1) My sister got the best appetizer (2) Kids' chicken salad at columbia harbour house (3) So many biscuits... (4) I plan to use it as a pencil holder on my desk. (5) Meeting cinderella before our dinner in 2011.

**(LBL)** (1) Lisa grilled wine country rabbit meatloaf baby portobello mushroom ragu and carrot puree nate wanted to like this dish more than he did (2) But great will soon get even better with the renovation of the elegant bistro de paris and expansion of the quick service bakery boulangerie patisserie (3) A nice selection of cheeses and accompaniments (4) This year we got a gorgeous reusable totebag (5) Meeting cinderella before our dinner in 2011.

Figure 3: Examples of sentence sequence retrieval for *NYC* (top) and *Disneyland* (bottom). In each set, we present a part of a query image stream, and its corresponding text output by our method and a baseline.

| Baselines | (GloMatch) | (CNN+LSTM) | (MLBL-B) | (RCN) | (RCN N>=8) |
|---|---|---|---|---|---|
| *NYC* | **92.7**% (139/150) | **80.0**% (120/150) | **69.3**% (104/150) | **54.0**% (81/150) | **57.0**% (131/230) |
| *Disneyland* | **95.3**% (143/150) | **82.0**% (123/150) | **70.7**% (106/150) | **56.0**% (84/150) | **60.1**% (143/238) |

Table 2: The results of AMT pairwise preference tests. We present the percentages of responses that turkers vote for our (CRCN) over baselines. The length of query streams is 5 except the last column, which has 8–10.

We first randomly sample 100 test streams from the two datasets. We first set the maximum number of images per query to 5. If a query is longer than that, we uniformly sample it to 5. In an AMT test, we show a query image stream $\mathcal{I}_q$, and a pair of passages generated by our method (CRCN) and one baseline in a random order. We ask turkers to choose more agreed text sequence with $\mathcal{I}_q$. We design the test as a pairwise comparison instead of a multiple-choice question to make answering and analysis easier. The questions look very similar to the examples of Fig.3. We obtain answers from three different turkers for each query. We compare with four baselines; we choose (MLBL-B) among the three variants of [12], and (CNN+LSTM) among mRNN-based methods. We also select (GloMatch), and (RCN) as the variants of our method.

Table 2 shows the results of AMT tests, which validate that AMT annotators prefer our results to those of baselines. The (GloMatch) is the worst because it uses too weak image representation (*i.e.* GIST and Tiny images). The differences between (CRCN) and (RCN) (*i.e.* 4th column of Table 2) are not as significant as previous quantitative measures, mainly because our query image stream is sampled to relatively short 5. The coherence becomes more critical as the passage is longer. To justify this argument, we run another set of AMT tests in which we use 8–10 images per query. As shown in the last column of Table 2, the performance margins between (CRCN) and (RCN) become larger as the lengths of query image streams increase. This result assures that as passages are longer, the coherence becomes more important, and thus (CRCN)'s output is more preferred by turkers.

# 5 Conclusion

We proposed an approach for retrieving sentence sequences for an image stream. We developed *coherent recurrent convolutional network* (CRCN), which consists of convolutional networks, bidirectional recurrent networks, and entity-based local coherence model. With quantitative evaluation and users studies using AMT on large collections of blog posts, we demonstrated that our CRCN approach outperformed other state-of-the-art candidate methods.

**Acknowledgements**. This research is partially supported by Hancom and Basic Science Research Program through National Research Foundation of Korea (2015R1C1A1A02036562).

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
