[Reviews · NeurIPS 2015]

Submitted by Assigned_Reviewer_1

This paper deals with a problem of retrieving natural sentences that describe a given sequence of images and forming a sequence of sentences. This problem is definitely significant in the research fields of computer vision and pattern recognition, but it is closely related to generating multiple sentences that describe given video content (e.g. [Rohrbach+ 2014]): Single video content can be regarded as a sequence of images, since video content can be separated into multiple video shots and every shot can be well described by a key frame. However in this paper, none in this line of researches have not been cited.

This paper is well written and easy to follow. The problem is well stated and solved with a set of solid techniques.

The proposed neural network model named CRCN acquires relationships between a sequence of natural sentences and a sequence of images. The architecture of the proposed model is technically sound and it also describes discourse relationships very well with the help of entity-based coherence models.

The method for generating a sequence of natural sentences is reasonable but rather primitive: Every natural sentence is directly associated with training images similar to a given image, and all the sequences are simply concatenated in the order of the corresponding images.

In Section 4.1, "We reuse the blog data of Disneyland from the dataset of [11], and newly collect the data of NYC, using the same crawling method with [11]," The data set has not been disclosed and the corresponding paper does not describe the details how to crawl the data set. This indicates that the authors of this paper is definitely almost the same as the ones in [11] and thus this situation deliberately lacks the anonymity.

Equation (2) might be incorrect, since it implies that s_t can be derived from only o_t and thus the network is not fully connected.
Summary: The proposed neural network model is technically sound and it also describes discourse relationships well with the help of entity-based coherence models. Meanwhile, the method for generating a sequence of sentences for a given image stream is rather ad-hoc. I think that this paper can be accepted as a poster paper as is.

Submitted by Assigned_Reviewer_2

The paper attacks the problem of describing a sequence of images from blog-posts with a sequence of consistent sentences. For this the paper proposes to first retrieve the K=5 most similar images and associated sentences from the training set for each query image. The main contribution of the paper lies in defining a way to select the most relevant sentences for the query image sequence, providing a coherent description. For this sentences are first embedded in a vector and then the sequence of sentences is modeled with a bidirectional LSTM. The output of the bi-directional LSTM is first fed through a relu and fully connected layer and then scored with a compatibility score between image and sentence. Additionally a local coherence model [1] is included to enforce the compatibility between sentences.

Strength / positive aspects: - The paper proposes a novel and effective architecture to retrieve coherent sentences for an image sequence. - The paper provides an extensive quantitative, qualitative, and human evaluation, showing the superiority of their approach against several baselines, not using coherence. They also provide an ablation experiment, removing the coherence model [1]. - The authors promise to release source code and dataset.

Weaknesses / Questions / Unclarities: 1. Line 94: The paper claims that there is "no mechanism for the coherence between sentences" in [5]. Although not the contribution of [5], [5] predicts an intermediate semantic representation of videos, which is coherent across sentences by modeling the topic of the multi-sentence description. 2. Correctness/clarity: Figure 2b does not seem to correspond to the description 3.3/Equation (2). While Figure 2b implies that the fully connected layer are connected to all sentences, this it not the case in Eq2, which implies that the parameters are shared across sentences, but only connected to the vector representing a single sentence. 3. A better metric to automatically evaluate the generated sentences is Meteor (http://www.cs.cmu.edu/~alavie/METEOR/) instead of BLEU, especially if there is only a single reference sentence. 4. Why two linear functions in Eq2 (W_{f2}, W_{f1}) are applied behind each other? Given that two linear functions are again a linear function the benefit is unclear. An ablation study showing the benefit of these functions would be interesting. 5. Why the same parameters of the fully connected layers are used for the BRNN output (o_t) and the local coherence model q (Equation 2)? 6. Is the paragraph vector [16] fine-tuned or kept fixed?

=== post rebuttal === After reading the rebuttal I recommend the paper for acceptance. The authors successfully addressed issues with the formulation, evaluation, and related work.

Please make the promised changes to the final and also clarify the following point in the final. 6. Is the paragraph vector [16] fine-tuned or kept fixed?
Summary: The paper proposes an interesting new model to retrieve coherent sentences for an image stream, which is convincingly evaluated. However, to be a convincing paper, several clarifications have to be made.

Submitted by Assigned_Reviewer_3

This work studies how to generate sentences from an image stream. It designs a

coherent recurrent convolutional network (CRCN), which consists of convolutional neural networks, bidirectional recurrent neural networks, and an entity-based local coherence model.

Overall it is a nice work, although it can be improved from the following aspects: * Several related work about video to sentence is missing, e.g.,

Jointly modeling deep video and compositional text to bridge vision and language in a unied framework

*While the quantitative results of the proposed method looks quite good, the user study in table 2 shows that it performs similar to one baseline, RCN. Significance test is needed to verify whether the improvement is reliable.
Summary: Nice algorithm for sentences generation from an image stream.

The quantitative results of the algo looks good but the user study only shows weak advantage over baselines.

Author Feedback
Author rebuttal: We thank reviewers for acknowledging the novelty of our problem, the appeal in the formulation, and convincing evaluation.

1. Retrieving instead of generating (R1,6)
Many previous sentence generation methods first build the vocabulary dictionary, from which a sentence is created by sequentially retrieving the highest scored words. Our approach has one-level higher granularity; we build the sentence dictionary, from which we make a passage by retrieving sequences. It is a main analogy that we use the same term 'generate', but following the reviewers' suggestion, we will replace the 'generate' with 'retrieve'.

2. Nonlinearity in fully connected layers (R1,3)
The reviews remark that the simple multiplication of the last two FC layers results in a single linear mapping. However, it was not, thanks to dropout. We assign 0.5 and 0.7 dropout rates to the two layers. Empirically, it improves generalization performance much over a single FC layer with dropout.
During rebuttal period, we also test the case of adding ReLu to the FC layers, but one key side-effect is that it severely suppresses the output of coherence model (i.e. g in Eq.(2) is quickly close to 0). Thus, the additional ReLu makes CRCN work like RCN. In NYC dataset, we obtain R@1, 5, 10, Med values as follows.
RCN 3.8, 18.3, 30.2, 29.0
CRCN+ReLU 6.9, 20.7, 33.3, 28.0
CRCN 11.7, 31.2, 43.6, 14.0

3. Evidence that CRCN is better with longer passage (R1,4)
Since the reviews' comment is very reasonable, we run new sets of AMT tests to check how the performance margins between RCN and CRCN vary according to the lengths of query image streams.
Pairwise preference of CRCN over RCN
5 images -> 8-10 images
NYC: 54.0% (81/150) -> 57.0% (131/230)
Disney: 56.0% (84/150) -> 60.1% (143/238)
This result supports our argument that as passages are longer, the coherence becomes more important, and thus CRCN's output is more preferred by general turkers.

4. Error in Eq.(2) (R2,3)
Thank you for the correction. The Eq.(2) should be
[S | g]= W_f2*W_f1*[O | q]
where O = [o_1|o_2| ... |o_N] , S = [s_1|s_2| ... |s_N].
We use the shared parameters for O and q, because we want the retrieval output to mix well the interaction between the content flows by BRNN and coherency. Our empirical results show this joint learning outperforms learning the two terms with separate parameters.

5. Better metrics instead of BLEU (R1,3)
Following reviews, we compute Cider and Meteor metrics. We observe the tendency of Cider and Meteor is similar to that of BLEU. Note that with retrieval metrics (R@K, MedRank), CRCN significantly outperforms RCN. For Disney dataset,
Baseline | Cider | Meteor
CNN+LSTM 10.0 4.51
CNN+RNN 0.4 1.34
MLBL-B 19.7 8.03
GloMatch 2.2 4.31
1NN 19.5 7.46
RCN 51.3 8.87
CRCN 52.7 8.78

6. Referring to video-sentence work (R2,3,4)
Multiple reviews suggest comparing with the video-sentence work (e.g. Rohrbach+2014, R. Xu+2015), which will be cited in the final draft.
Our key novelty is that we explicitly include the coherence model, which is more critical for image streams than videos. Unlike videos, consecutive images in streams may show sharp changes of visual contents, which cause the abrupt discontinuity between contents of consecutive sentences. Thus the coherence model is more demanded to make output passages fluent.

7. Anonymity (R2)
As R2 pointed out, the dataset of [11] is not publicly available. But the authors of [11] were open to share their dataset upon request.

8. Why CRCN better than RCN in retrieval? (R1,3)
CRCN is only slightly better than RCN in language metrics but significantly better in retrieval metrics. It means that RCN is OK with retrieving fairly good solutions, but not good at ranking the only correct solution high compared to CRCN.
The coherence model crates descriptors that capture various aspects regarding coherency, and the interaction between this descriptor and BRNN output are jointly learned via two FC layers. As R1 mentioned, the test text streams to be ranked are already coherent in some sense (because they are passages written by human), but our model does not simply summate the content and coherence terms, but learns the complex interactions between them; such modeling power can help pinpoint the correct retrieval.

9. Difference from Karpathy's [9] (R1)
R1's comments are correct. Plus, we add three more differences. First, our method includes the coherent model for smooth sentence transitions. Second, the final retrieval output of [9] is existing sentences in the training set, whereas our output passages do not exist in the training set. Third, by nature of our problem, the compatibility of sequential ordering is more important than [9] that parses the image into multiple semantic regions, and measures their fitness with words of a sentence with rather free ordering.

All the other comments are about the details of algorithms and experiments, which will be resolved in the final draft.